# Glass-to-Glass Fusion Bonding Quality and Strength Evaluation with Time, Applied Force, and Heat

**DOI:** 10.3390/mi13111892

**Published:** 2022-11-02

**Authors:** Nhi N. Trinh, Leslie A. Simms, Bradley S. Chew, Alexander Weinstein, Valeria La Saponara, Mitchell M. McCartney, Nicholas J. Kenyon, Cristina E. Davis

**Affiliations:** 1Department of Biomedical Engineering, One Shields Avenue, University of California Davis, Davis, CA 95616, USA; 2Department of Mechanical and Aerospace Engineering, One Shields Avenue, University of California Davis, Davis, CA 95616, USA; 3UC Davis Lung Center, One Shields Avenue, University of California Davis, Davis, CA 95616, USA; 4VA Northern California Health Care System, 10535 Hospital Way, Mather, CA 95655, USA; 5Department of Internal Medicine, 4150 V Street, University of California Davis, Davis, CA 95616, USA

**Keywords:** wafer bonding, fusion bonding, Borofloat glass, plasma activation

## Abstract

A bonding process was developed for glass-to-glass fusion bonding using Borofloat 33 wafers, resulting in high bonding yield and high flexural strength. The Borofloat 33 wafers went through a two-step process with a pre-bond and high-temperature bond in a furnace. The pre-bond process included surface activation bonding using O_2_ plasma and N_2_ microwave (MW) radical activation, where the glass wafers were brought into contact in a vacuum environment in an EVG 501 Wafer Bonder. The optimal hold time in the EVG 501 Wafer bonder was investigated and concluded to be a 3 h hold time. The bonding parameters in the furnace were investigated for hold time, applied force, and high bonding temperature. It was concluded that the optimal parameters for glass-to-glass Borofloat 33 wafer bonding were at 550 °C with a hold time of 1 h with 550 N of applied force.

## 1. Introduction

The global market for micro-electrical mechanical systems (MEMSs) and sensors has continued to rise where the technical requirements of MEMS-based substrates have expanded to include higher biological compatibility, optical transparency, corrosion and heat resistance, and a need to withstand high transition temperatures. Using glass as a primary substrate in MEMS devices has increased in popularity due to its unique physical properties that enable MEMS devices to meet these requirements and more. Glass substrates, such as Borofloat 33, have many uses in scientific and industrial areas, such as microelectronics, biotechnology, optics, and chemistry. Characterized by its similar optical properties to fused silica, it is less expensive with high resistance to thermal shock (CTE = $3.25\times 10^{-6}\;K^{-1}$) and chemicals, such as alkalis, acids, and organic substances [1]. 

Various methods have been explored to bond glass wafers, such as anodic bonding, where a hermetic connection is created between the glass to glass or glass with an intermediate layer. With an intermediate layer, the substrates are heated to 400 °C with a high voltage applied up to 1 kV. At higher temperatures, the sodium oxide splits into sodium and oxygen ions where the intermediate layer acts as a diffusion barrier. The sodium ions migrate towards the cathode, whereas the oxygen ions migrate towards the anode, creating an interface at the intermediate layer [2,3,4,5,6]. However, the use of an intermediate layer may be unfavorable for the optical transparency of devices and certain anodic bonding equipment may not be readily available. Other processes have been developed to produce strong bonding strength at lower temperatures, even without cleanroom facilities. By prewashing the wafers with acetone, substrates can be bonded by the continuous flow of demineralized water. However, fabricating strong bonding strength of glass-to-glass wafers at low temperatures and loads has not been obtained [7].

Experiments have been conducted to determine effective bonding procedures; however, the influence of cleaning procedures prior to the bonding can affect the quality of the bond. Contamination of particulates can directly affect the quality of the bond and bond strength by acting as spacers between the wafer surfaces. Cleaning methods for fusion bonding for microfluidic devices include various wet chemical baths. Lin et al. and Liao et al. used an acetone and piranha solution (H_2_SO_4_:H_2_O_2_ = 3:1), followed by a rinse with deionized water and wafers were blown with N_2_ [8,9]. Other chemicals were used in addition to piranha solution, such as hydrochloric acid and ammonium hydroxide solution in the case of Plößl et al. [10]. Additional chemical solutions also include KOH, HNO_3_, and ethyl alcohol [11,12,13,14]. Based on all the chemical solutions, piranha and HCl solutions are commonly used to minimize organic contamination [15].

Separate explorations have been made into surface-activated bonding (SAB) methods, by treating the surface of the substrate with various ions or radicals. In SAB, wafers are directly bonded after the sample surface is activated using reactive ion etching (RIE), inductively coupled plasma, ozone, and UV radiation plasma carried in different atmospheres. Furthermore, room temperature bonding is possible with the SAB method. Notably, N_2_ and O_2_ have been used in a two-step plasma surface-activation process where the O_2_ plasma removes contaminants and reactivates native oxides on the bond interface [16,17]. Wang et al. proposed a bonding mechanism with surface activation. During O_2_ plasma activation, oxygen free radicals break the Si-O-Si and Si-Si covalent bonds. The chemical bonds are regenerated into Si-O covalent bonds due to the high energy of oxygen free radicals. The newly generated Si-O bonds can create a dehydration reaction to form covalent bonds of Si-O-Si where the bonding of substrates can be achieved [18]. Subsequent N_2_ microwave (MW) radical activation can further generate chemical reactive surfaces and decreasing the surface roughness of oxide substrates without damaging the nanostructured surfaces [18,19,20]. SAB has generally been utilized for various materials, such as silicon, silicon carbide, and aluminum oxide, whereas the use of SAB with glass has only been explored with the use of an oxide intermediate layer [21,22,23,24]. Nevertheless, SAB may not always generate high-quality bonded glass wafers due to the void formation of trapped air between the wafers as they make contact. Moreover, the surface energy of the bond interface deteriorates exponentially over time.

Alternatively, fusion bonding proves to be a common method of bonding glass substrates, as it is an inexpensive and versatile way of creating a hermetic seal [15]; a good bonding strength with a high bonding yield can be achieved. Numerous studies have utilized fusion bonding, for example, a glass-to-glass fusion bonding study that used a buffered oxide etch (BOE) and pre-heating a coverslip in a furnace at 400 °C before heating up to 580 °C [8]. Other pre-bonding treatments have been used, such as a two-step baking process with a dip into hydrochloride solution, followed by a BOE bath where the substrates are bonded at 580 °C for 20 min [25]. A bench-scale press furnace has also been used to bond the glass substrates for 6 h at 590 °C and apply a 12 PSI pressure [12]. Other fusion bonding studies used 630 °C and 1.98 lb of applied force, or annealing wafers at 1100 °C [9,26]. Varying parameters were proposed for a multitude of various glass materials, such as Pyrex 7740, borosilicate glass coverslip, fused silica wafers, and Thermo Scientific soda lime glass slides, all of which have varying thicknesses [8,9,12,15,25,26]. Although there have been numerous successes with glass-to-glass bonding at high bonding temperatures, there is a need to optimize the parameters of bonding temperature, hold times, and applied load to bond glass substrates, in this case, with Borofloat 33. 

A fusion bond is a settling bond where a chemical reaction with van der Waals hydroxyl bonds is transferred to covalent siloxane bonds. This process can occur in ambient conditions and can be accelerated, strengthened, and enhanced with annealing. The chemical reaction between the hydroxyl molecules on the mating surfaces is responsible for the bonding of glass wafers. If the bond becomes monolithic, the bond is as strong as the interface in the bulk of the material. However, to have a monolithic fusion bond, a severe annealing procedure has to be executed to prevent thermal shock through such means as having a gradual heating and cooling cycle [27]. 

When examining the thermal properties of glass wafers for bonding, the viscosity and annealing of the glass are key factors for a product able to retain its shape after forming with little to no crack formation. Some of the key parameters affecting viscosity are the glass transition temperature (T_g_), the softening point, and the annealing point. The T_g_ of borosilicate glass, generally around 500–550 °C, depends on the molar ratio of silica to boron. The softening point, around 820 °C in the case of borosilicate glass, is where the glass starts to deform under its own weight at that viscosity. Finally, the annealing point is defined to be the temperature at which the stresses generated during the processing can be released by viscous relaxation. To release the internal stresses, the glass needs to be heated above the annealing point and subsequently cooled down slowly [28]. For fusion bonding, quartz substrates are bonded above 1000 °C and borosilicate glass is bonded between 600 and 800 °C. Based on the thermal properties of borosilicate glass, the ideal bonding temperature would range between T_g_ and the softening point of glass, in the 500–820 °C range [28]. 

In this study, a bonding procedure for Borofloat 33 glass was developed, which incorporated two major steps of a pre-bond step and a high-temperature bond step. The pre-bond step utilizes SAB, where the wafers are in contact in a vacuum setting to generate a temporary bond and alleviate possible contamination. By including SAB in this pre-bond step, the formation of Newton’s rings and voids was greatly reduced. During the pre-bond step, the wafers are heated up to 450 °C with 500 N of applied force and the time duration was investigated. 

For the high-temperature bond step, the wafers will be evaluated based on the bonding percentage of the wafer and the flexural strength test. The bonding percentage of the wafer would be obtained via ImageJ analysis, where a threshold is applied to assess the unbonded regions of the wafer. As for the flexural strength tests, the use of bending tests instead of peel tests is utilized as a repeatable assessment of adhesion strength, with numerous studies conducted on bending displacement tests, four-point bending tests, or three-point bending tests [29,30,31]. Based on these variables, the high-temperature bond step can be evaluated and investigated based on bonding temperature, hold time, and applied force. 

## 2. Materials and Methods

### 2.1. Wafer Preparation

Figure 1 describes the full process of fusion bonding the borosilicate glass wafers. The borosilicate glass wafers used in this experiment were Borofloat 33 glass wafers with a diameter of (100 ± 0.1) mm and thickness of 700 µm (University Wafer, South Boston, MA, USA). The topside and backside surface roughness of the wafers are both <1 nm, with a warp of <30 µm. The bonding process was conducted in an ISO 5, Class 100 cleanroom environment. All wafers were cleaned in a piranha solution (H_2_SO_4_:H_2_O_2_ = 4:1) that was prepared at ambient temperature and were cleaned for 10 min (Figure 1a). All samples were immediately cleaned after the preparation of the piranha solution. The wafers were then rinsed with deionized (DI) water and dried in pure N_2_ in a Semitool Spin Rinser (Semitool PSC-101, Semitool, Kalispell, MT, USA). The wafers were prepared to be used immediately, to ensure that all organic surface contaminants were removed. 

### 2.2. Two-Step Plasma Activation

The surface of both wafers underwent a two-step plasma surface activation process in the EVG 810 Plasma Activation (EVG, St Florian, Austria) (Figure 1b). The surface was first activated with an O_2_ plasma at 75 W/100 W (low-frequency power/high-frequency power) for 20 s with a gas flow of 2500 sccm, followed by N_2_ MW radicals at 75 W/150 W (low-frequency power/high-frequency power) for 20 s with a gas flow of 5000 sccm. The wafers were then aligned using the EVG 620 Mask Aligner (EVG, St Florian, Austria) and EVG 4 in/100 mm Bond Chuck (EVG, St Florian, Austria), allowing for a higher degree of alignment accuracy. To avoid direct contact between the wafers prior to the pre-bond soak, the 4 in/100 mm bond chuck comes with complimentary separation wafer flags as part of the stock EVG bonder configuration to be set between the wafers as spacers. The full wafer stack is held down using the wafer clamping glass and retention tabs from the bond chuck. On top of the wafer stack, a 4 in/100 mm Graphite Foil 106.5x1 and a 4 in/100 mm Graphite Electrode without a center for a bow pin (EVG, St Florian, Austria) are installed. 

We evaluated whether the surface activation process is necessary to the bonding process. Wafers were tested with and without the two-step plasma activation step, then went through the entire fusion bonding process.

### 2.3. Pre-Bond Soak

From the EVG 620 Mask Aligner, the bond chuck is transferred to the chamber of the EVG 501 Wafer Bonder (EVG, St Florian, Austria) for a pre-bond soak (Figure 1c). This chamber enables the wafer assembly to be in a vacuum environment that is then heated to 450 °C at a rate of 30 °C/min followed by a 10 min period for even heat distribution. The flags are then removed to allow the wafers to have direct contact where 500 N of load is applied at a rate of 100 N/min. The EVG 501 has a flat plate and piston where pressure is applied to have intimate contact between the glass wafers to form chemical bonds. This environment is held constant and three duration periods were tested (3 h, 5 h, and 9 h hold), before gradually returning to ambient temperature.

### 2.4. Glass-to-Glass Bonding Process

After the pre-bond soak, the bonded pair is assembled into a custom-built screw fixture. The fixture consists of 410 stainless-steel plates, 4 in/100 mm graphite discs, bolts, and screws. The 4 in/100 mm graphite discs were installed at the interface between the glass wafers and 410 stainless steel plates due to the difference in thermal expansion coefficients to avoid fracture propagation during the high-temperature step. The stainless-steel plates were machined to have holes for the bolts and were finished with electropolishing to provide a smoother surface finish. The plates were coated with a 200 nm layer of silicon nitride (SiNx) using a PlasmaTherm Vision 310 PECVD (Plasma-Therm, St. Petersburg, FL, USA). 

The wafer stack from the EVG 501 Wafer Bonder was transferred and assembled to the fixture (Figure 1d), applying torque to the hex nuts to achieve a prescribed load. Three loads were tested (550 N, 840 N, and 1110 N). The torque to apply is calculated through Equation (1): (1)T=k∗d∗F
where T is the tightening torque (N·m), k is the torque coefficient (k=0.2 for stainless steel), d is the screw nominal diameter (d=9.525 mm), and F is the axial force (N) [32].

A programmable furnace (Vulcan-Hart 3-550, ESP Chemicals, Tucson, AZ, USA) was used to fusion bond the borosilicate glass together at higher temperatures. Three temperatures were tested (550 °C, 650 °C, and 700 °C) at a fixed rate of 4 °C/min. The furnace is held at a high temperature and three holding periods were tested (1 h, 2 h, and 4 h), before allowing the bonded wafers to gradually cool down to room temperature (Figure 1e). 

For all bonding conditions, three wafers were tested with each bonding variable. The bonding variables to be assessed are bonding temperature (550 °C, 650 °C, and 700 °C), hold times (1 h, 2 h, and 4 h), and applied load (550 N, 840 N, and 1110 N). With all wafers going to the glass-to-glass bonding process, the pre-bond soak time in the EVG 501 Wafer Bonder was 3 h at 450 °C with 500 N applied force. 

### 2.5. Data Analysis

The bonded wafers are visually assessed for the percentage of the wafer that has bonded. This was evaluated using ImageJ analysis software version 1.53f51 (NIH, Bethesda, MD, USA), to quantify the number and size of voids in each sample. Photos of the wafers were taken against a white background in a lightbox where a threshold ranging from 20 to 40% was applied on ImageJ to create a contrast between the voids and bonded regions. The voids and unbonded regions were then visually assessed and evaluated once the threshold was applied. The area of the wafer and unbonded regions was measured to assess the bonding percentage. 

To be reliably integrated into devices, bonded wafers need to have sufficient resistance to bending and exhibit minimal warping, two metrics that will increase the likelihood of successful bonding [33]. A review of destructive methods to assess the bonding strength in wafers is given by Vallin et al. [31]: double cantilever beam tests, peel/tensile tests, blister tests, and chevron tests. Double cantilever beam tests are based on the insertion of a thin blade into the bond and the measurement of the crack propagating from the initial discontinuity. The tests are very sensitive to the measurement of this crack length, [31]. In tensile tests, bonded samples are glued onto studs, which are loaded in tension. For brittle materials, these tests are sensitive to misaligned loads and, in these circumstances, edge effects will cause an early fracture; moreover, a large scatter has also been reported [31]. With blister tests, debonding is caused by the creation of a cavity below the film, with the cavity being pressurized. Although the tests have been successful for some thin-film systems, there are problems, such as the compliance of the testing system and the potential of stress-corrosion cracking between the debond and the pressurized environment [34]. In the chevron tests, a specially formed notch is introduced. The crack, which needs to be measured, initially grows in a stable manner, followed by unstable propagation. Dauskardt et al. proposed the use of bending tests in place of peel tests, as a more robust and repeatable assessment of adhesion strength in multi-layer thin-film structures [34]. The quality of bonding in MEMS devices can be assessed through bending tests [30,33,34,35,36,37,38,39,40], from the amount of bending displacement (e.g., Ikeda et al.; Tanaka et al.; Djuzhev et al.), while some authors have carried out four-point bending tests to compute the critical adhesion energy (Dauskardt et al.; Kwon et al.; Zou et al., etc.) or three-point bending tests (Kalkowski et al.). In our work, bonding strength quality is assessed through a modified version of the ASTM C1161-18 standard, “Standard Test Method for Flexural Strength of Advanced Ceramics at Ambient Temperature”. The flexural strength tests performed could not conform completely to the standard (test specimens may be 3 × 4 × 45 to 50 mm^3^ in size) due to the geometry of the samples. A rolling, non-articulating three-point fixture configuration was used, with a span length of 39.56 mm, and the corresponding displacement rate was computed based on the required strain rate. Autoclave adhesive covered by its non-sticky paper was applied on the rollers for improved support of the samples, based on the setup of ASTM C1505-15, “Standard Test Method of Breaking Strength and Modulus of Rupture of Ceramic Tiles and Glass Tiles by Three-Point Loading”, which was also consulted.

The samples were tested until failure using an instrumented MTS 810 hydraulic testing machine, at a displacement rate of 1.11 mm/min computed based on support span (L = 39.56 mm), sample thickness (t = 1.40 mm), and desired strain rate (10^−4^ s^−1^). The material ultimate flexural strength S was calculated using Equation (2) (ASTM C1161-18):(2)$$S=\frac{3PL}{2bt^{2}}$$
where P is the maximum compressive load (break force), L is the support span, b is the specimen width, and t is the specimen thickness. This formula should be considered an estimate because of the deviation from the standard sample geometry, but the testing procedure was the same for all samples. Examples of how glass substrates are evaluated for flexural strength are in Figure 2. Statistical analyses were performed using MATLAB version R2021a (MathWorks, MA, USA).

Statistical analysis was conducted using a Wilcoxon rank sum test to calculate *p*-values comparing each treatment of the bonding parameters.

## 3. Results

### 3.1. Surface-Activation Evaluation

The two-step plasma surface-activation process in Figure 1b was evaluated to verify if the process was vital to the bonding process, particularly the pre-bond soak. The fixed variables were a 5 h EVG 501 Wafer Bonder hold at 400 °C. During the furnace bonding process, the samples were bonded at 650 °C, 550 N of applied load with a 2 h hold time. As shown in Figure 3, the ultimate flexural strength for tests with SAB treatment was (77.10 ± 14.50) MPa and (79.80 ± 16.00) MPa without SAB treatment, that is, the two groups’ flexural strengths were statistically equivalent. However, a bonding process with surface activation produces wafers have slightly higher percentages (88.40 ± 1.57)% for those without SAB and (93.86 ± 0.75)% (see Appendix A for Wilcoxon rank sum tests). With the use of plasma activation, the O_2_ plasma removes surface contaminants, such as hydrocarbons and particles, where a hydrophilic oxide layer can be adsorbed by exposed hydroxyl groups. The N_2_ radicals in the second step of the plasma treatment were then used to further remove surface contaminants, increasing the level of contact between the glass wafers to create chemical bonds. At the contact surface, the chemical reaction between hydroxyl molecules is responsible for the bonding of glass-to-glass wafers where plasma surface treatment leads to a higher bonding percentage and higher quality bond.

### 3.2. Pre-Bond Soak Bond Percentage Evaluation

The pre-bond soak (Figure 1c) was performed at 450 °C, over a range of hold times (3 h, 5 h, and 9 h), to evaluate the optimal time to expose the wafer assembly to a vacuum environment as the wafers make contact. As the glass transition temperature is T_g_ = 525 °C, the pre-bond soak would produce relatively weak bonding. In order to anneal the glass wafers, higher temperatures are required. However, this pre-bond soak step was introduced to allow the wafer samples to have contact in a vacuum environment after undergoing surface plasma activation, to minimize voids being created. Based on previous attempts, voids can easily be created during wafer assembly and after the surface plasma activation process, even in a cleanroom environment, as particles could contaminate the surface. The hold times were evaluated to see the effect on the percentage of the wafer bonding time (3 h, 5 h, and 9 h), with n=3 samples per holding time. Results are shown in Table 1. The 3 h bonding time had the highest percentage of the wafer bonded and had the least amount of standard error (93.24 ± 0.43)% relative to other hold times. Wilcoxon rank sum tests were conducted between 3 and 5 h, 5 and 9 h, and 3 and 9 h where all values were *p* > 0.05 (*p* = 0.70, *p* = 1.00, and *p* = 0.70, respectively). This indicates that the data give little confidence to conclude that the overall means are statistically different from each other. With a longer holding time, the wafers could experience distortion, as thermally induced stress can be introduced in the stack as the chemical bonds formed in the pre-bond soak are electrostatic and are not permanent. It should be noted that there could also be thermal stress due to the difference in ambient and holding temperature. From the results, a pre-bond soak time of 3 h produces a partially bonded wafer with the least amount of scattering, as the wafers continue to the high-temperature bonding step. 

### 3.3. Furnace Bonding Variables

The bonding variables are critical parameters for the furnace bonding process. The optimum bonding variables were investigated based on hold time, applied load, and high temperature. For hold times, the fixed variables were at 650 °C for the high temperature and 500 N of applied load. From Figure 4, there were varying results based on trying to achieve a high bonding percentage and bonding strength bonded wafer. Based on the bonding percentage, the longer hold times produced wafers that had a higher bonding percentage for 4 h at (95.82 ± 0.75)% but had an inverse relationship with bonding strength where the ultimate flexural strength was (119.60 ± 8.90) MPa (see Appendix A for Wilcoxon rank sum tests). Exposing the wafer assembly to a high bonding temperature for a longer period of time allows the glass substrates to interact with each other more to anneal and create a hermetic seal. However, the longer hold time can thermally induce stress to the wafers. Therefore, based on Figure 4, the optimal hold time is 1 h.

As for the applied load, the fixed variables were at 650 °C for the high temperature and 2 h of hold time. From the results in Figure 5, the bonding percentages differed slightly from each other; it was evident that applying 550 N of force yielded the best results. By applying 550 N to the wafer samples, (93.04 ± 2.05)% of the wafer was bonded, with an ultimate flexural strength of (139.97 ± 7.81) MPa (see Appendix A for Wilcoxon rank sum tests). Applying forces to the wafer samples causes them to have direct contact and adhere with each other as they reach the glass transition phase. However, at higher force values, stress concentrations can be created around areas of imperfections on the wafer from when it was first manufactured or increase in thermally induced stress. As a result, the optimal applied force with the screw fixture is 550 N.

When evaluating the high bonding temperature, the fixed variables were at a 2 h hold with a 550 N force applied. Figure 6 shows that with bonding percentage, 550 °C and 700 °C treatments were close in value at (97.03 ± 0.30)% and (97.71 ± 0.27)%, respectively. However, data for the flexural strength showed otherwise, where the strength of the 700 °C was significantly lower than the 550 °C, with (72.56 ± 14.50) MPa and (102.0 ± 20.51) MPa, respectively (see Appendix A for Wilcoxon rank sum tests). Since the 700 °C temperature is higher than the T_g_ of Borofloat 33, this value is closer to the softening point of 820 °C, where the glass would start to deform under its own weight. Such results for the 700 °C treatment indicate that the treatment at 550 °C is superior.

For all bonding parameters, the bond evaluation was conducted based on the bonding percentage of the wafer and the flexural strength of the bond. However, other modes of bond evaluation could be conducted, such as atomic force microscopy (AFM), scanning electron microscopy (SEM), or tensile strength tests. There were limitations with access to such equipment and fixtures to conduct additional bond evaluation modes that should be considered going forward with this bonding process. Further testing was conducted on the external surface roughness after the bonding procedure to evaluate the finish of the glass wafers (see Appendix A) The use of this bonding process is attractive for an application that requires additional deposition of material on the external surface.

## 4. Conclusions

Through this study, a fusion bonding process was developed for 4 in/100 mm, 700 µm Borofloat 33 with a pre-bond soak and a high-temperature bonding step. For the high-temperature bonding step, the optimal conditions for direct fusion bonding were assessed by varying high bonding temperature, hold time, and applied force. The pre-bond soak process included a two-step surface activation step and the EVG501 Wafer Bonder. In the EVG501 Wafer Bonder, the bonding percentages showed that a hold time of 3 h at 450 °C yielded the best results. As for the high-temperature bonding step, based on the bonding percentages and flexural strength calculated, it was found that, among the tested conditions, Borofloat 33 4 in wafers bonded most optimally at 550 °C, with a hold time of 1 h and 550 N of applied force. This bonding process yielded samples that had a high bonding percentage and high flexural strength. 

## Figures and Tables

**Figure 1 micromachines-13-01892-f001:**
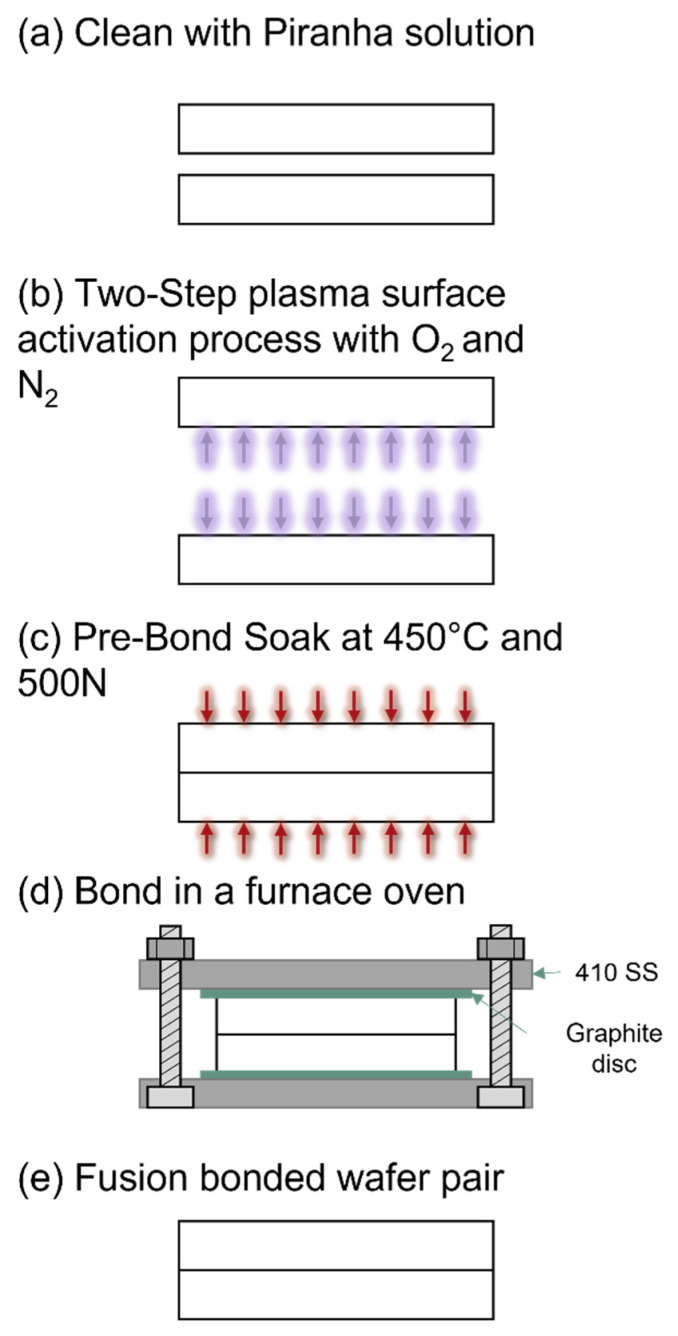
Schematic of the process of fusion bonding borosilicate glass. (**a**) Cleaning with piranha solution to remove all organic materials from two separate borosilicate wafers. (**b**) Using a two-step plasma surface activation process with O_2_ plasma and N_2_ MW radical activation. (**c**) Pre-bond soak at 450 °C with 500 N applied force. (**d**) Fusion bond substrates with a custom screw fixture to apply force in a furnace. (**e**) Bonded Borofloat 33 wafers.

**Figure 2 micromachines-13-01892-f002:**
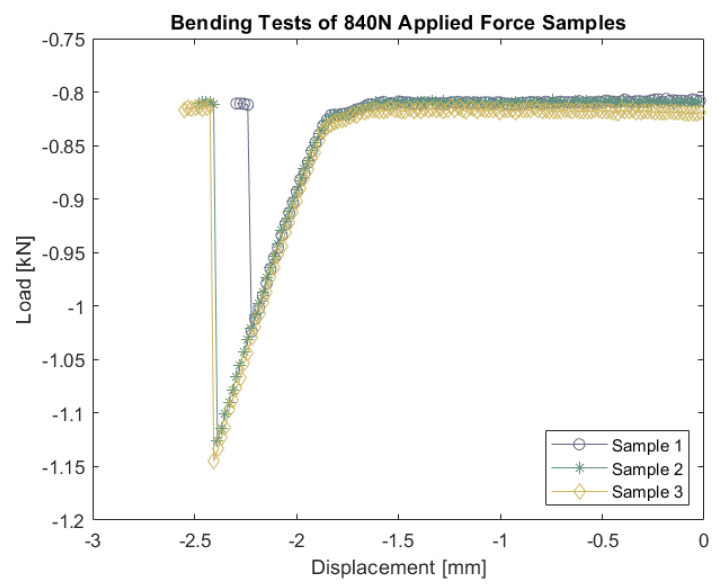
Displacement versus load plots measured during bending tests, conducted on *n* = 3 samples; the flexural strength of the sample was calculated based on the average of the break forces.

**Figure 3 micromachines-13-01892-f003:**
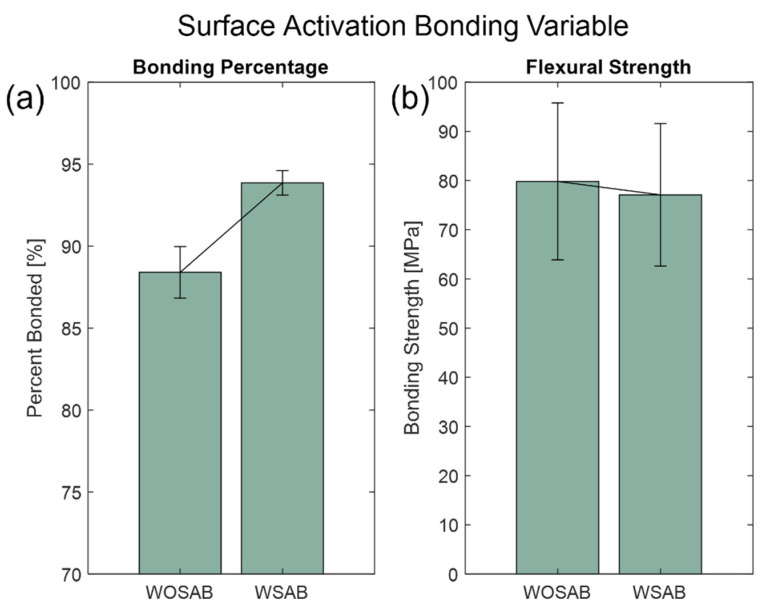
Surface-activation bonding variable with SAB (WSAB) and without SAB (WOSAB) show (**a**) the average bonding percentage and (**b**) flexural strength with sample standard error bars.

**Figure 4 micromachines-13-01892-f004:**
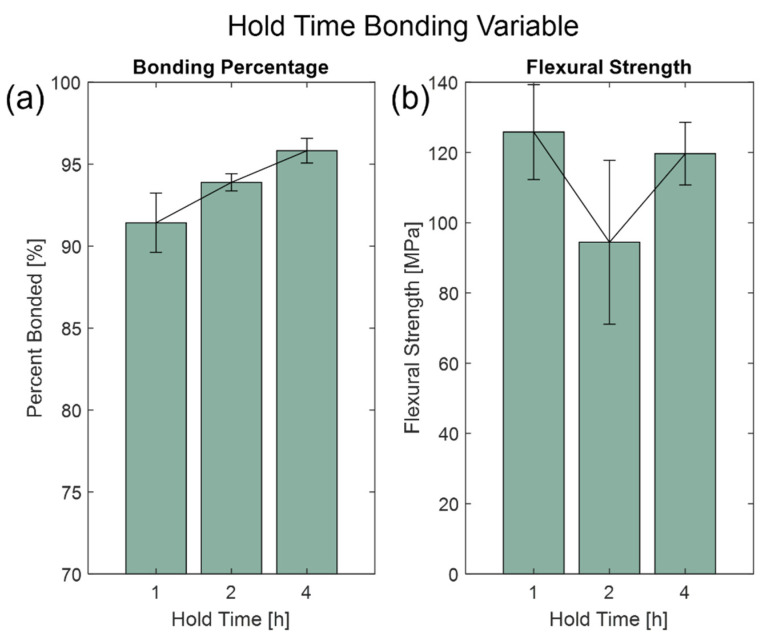
Hold time bonding variable at 1 h, 2 h, and 4 h. Bar graphs show (**a**) the average bonding percentages and (**b**) flexural strength with sample standard error bars.

**Figure 5 micromachines-13-01892-f005:**
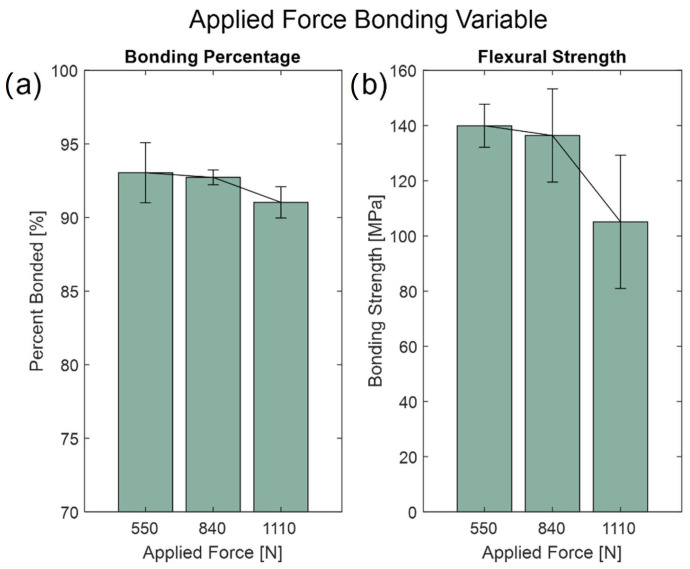
Applied force bonding variables at 550 N, 840 N, and 1110 N show (**a**) the average bonding percentage and (**b**) flexural strength with sample standard error bars.

**Figure 6 micromachines-13-01892-f006:**
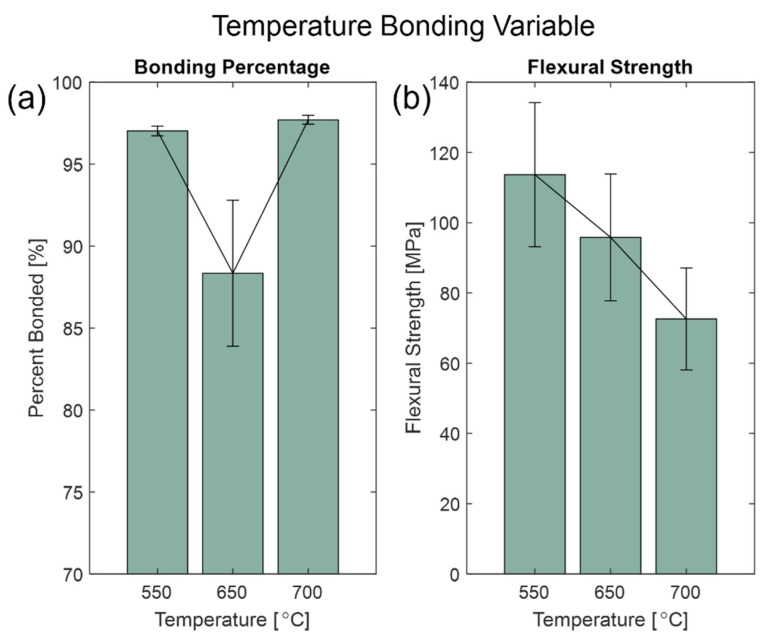
High-temperature bonding variable at 550 °C, 650 °C, and 700 °C. Bar graphs show (**a**) the average bonding percentage and (**b**) flexural strength with sample standard error bars.

**Table 1 micromachines-13-01892-t001:** Results of pre-bond soak percentage bonded. N = 3 experiments were conducted for each of the EVG hold times. Sample standard error describes variability across multiple wafer samples.

Trial	3 h (%)	5 h (%)	9 h (%)
1	92.46	91.57	92.25
2	93.33	89.76	95.19
3	93.93	94.15	84.71
Average	93.24	91.83	90.72
Sample Std Error	0.43	1.28	3.12

## Data Availability

Research data are available from the authors.

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
