# Peer review of "Glass-to-Glass Fusion Bonding Quality and Strength Evaluation with Time, Applied Force, and Heat"

_micromachines, 2022, doi:10.3390/mi13111892_

Round 1

Reviewer 1 Report

1. In the introduction, it is necessary to analyze the methods for evaluating the bond between wafers. It is important to consider not only qualitative methods of bond evaluation, but also quantitative ones. There are destructive and non-destructive methods of interfacial voids detection. Show that the accuracy of one or another method depends on the size of the bubbles to be detected detection, the properties of bonding materials, the stability of the interaction between layers, etc. 

2. The Graphite Foil and Graphite Electrode dimensions should be converted into SI units. 

3. Instead of "30 °C/min." it should be written "30 °C/min"  (line 147).

4. Specify the dimension for screw nominal diameter ? in equation (1).

5. The quality of bonded wafers should be characterized by the unbonded area. The bonding percentage as a characteristic is not clear and does not indicate the physical nature of the bonding.  

6. How was the threshold chosen to differentiate between voids and bonded regions while calculating the bonding percentage in ImageJ software?

7. Specimen thickness (line 216) and the screw nominal diameter (line 165) are marked using the same symbol d. It is necessary to denote them by different symbols.

8. Flexural strength characterizes the strength of a specimen of bonded wafers. However, flexural strength cannot characterize bond strength. 

9. Explain the negative load in the flexural test (Fig. 2 b).

10. What was the size of the observed area used to determine the bonding percentage? How were the bonding percentages obtained with such accuracy?

11. Why is equation (1) presented if the calculations are not given? 

12. An overview of the methods given in references [22-28] should be presented in the Introduction section.

Reviewer 2 Report

See attachment for my feedback

Reviewer 3 Report

This manuscript describes a procedure for the fusion bonding of glass to glass. The fusing bonding is not new, and the process in this manuscript does not have an improvement. The authors did not explicitly mention the novelty of the method.

Fusion bonding can be done outside of the cleanroom. See the article of Lapidus group: https://www.mdpi.com/2072-666X/8/1/16

The authors should discuss their methods and other methods, as shown in the abovementioned reference.

The surface roughness measured from Dektak does not make sense as Dektak has a limitation on a micro-range measurement, generally used for characterized photoresist thickness (the stylus size is already in the range of 5 microns). The authors should address this comment in the manuscript.

Round 2

Reviewer 1 Report

The authors of the paper have answered all comments except one (No 1). 

It should be noted that there is no analysis of methods for determining bond strength.

The authors of the paper added a paragraph (lines 125-132). However, they did not cite any literature references. This needs to be corrected. There are more such methods. See, for example, the review of Vallin, Ö., Jonsson, K.,  Lindberg, U. (2005). Adhesion quantification methods for wafer bonding. Materials Science and Engineering: R: Reports, 50(4-5), 109-165. 

In the introduction, it is necessary to analyze the methods for evaluating the bond between wafers. It is important to consider not only qualitative methods of bond evaluation, but also quantitative (double cantilever beam, tensile, chevron, blister, three-point bending and four-point bending tests) ones.

It is necessary to explain the selection of bonding area evaluation method for bond quality analysis. The bond area is not as accurate in evaluating bond quality compared to the bending test methods of specimens, specifically the three-point bending test. Moreover, the authors perform bend tests in accordance with ASTM C1505-15 on an MTS 810 hydraulic testing machine. It was possible to determine the bond strength in the bending testing of specimens.

It is not clear what limitations are written concerning the tensile test, if the MTS 810 hydraulic testing machine allows performing tensile testing (lines 342-350). Apparently, it should be referred to the difficulty of preparing specimens, rather than the lack of testing equipment. 

Reviewer 2 Report

The authors have considered the feedback, and accordingly revised the manuscript. Major improvements have been made, the revised work is worth publishing. 

Only some minor remarks:

* in the response letter various additional references are listed, but not all are included in the manuscript (e.g. Oosterbroek; Tiggelaar). I suggest to include these 2 as well, since they discuss also substrate cleaning prior to bonding (as well as various in anneal parameters).

* the authors mention in their response that they prepare Piranha solution at ambient temperature but not measure the solution temperature during use. I suggest to include in the revision that Piranha solution is prepared at ambient/room temperature, but that the temperature during use is not measured (it will certainly not be ambient temp, due to the exothermic nature of the reaction). This to avoid confusion for readers.

* the authors use graphite foil between the 410 SS clamping tool and the BF33 stack, to avoid fracturing of the stack during annealing (this is explained in the response to my feedback). I suggest to include this reasoning in the revised manuscript, incl. that without use of graphite foil the stack will fracture during annealing (as occured during not shown/discussed preliminary experiments). This will benefit clarity.

* Is the applied force during pre-bond soak 500 N (as mentioned in the caption of Fig. 1), or 550N (as mentioned in Fig. 1 itself + subsection 2.3)? Please clarify.

Reviewer 3 Report

The authors did not thoroughly address my comments: comment number 2 and 3.

Some of the authors' answers have many (English) grammar errors. I suggest a revision.

Round 3

Reviewer 1 Report

Comments on the analysis of bond strength methods have been corrected. I wish the authors of the article in further studies to progress from qualitative methods of studying bonding strength to quantitative ones, since corresponding laboratory equipment and experience in performing highly accurate and difficult testing are available.

Reviewer 3 Report

I do not have further comments.